# COVID-19 in Pregnancy: Knowledge about the Vaccine and the Effect of the Virus. Reliability and Results of the MAMA-19 Questionnaire

**DOI:** 10.3390/ijerph192214886

**Published:** 2022-11-12

**Authors:** Alice Mannocci, Claudia Scaglione, Giovanna Casella, Antonio Lanzone, Giuseppe La Torre

**Affiliations:** 1Faculty of Economics, Universitas Mercatorum, Piazza Mattei 10, 00186 Rome, Italy; 2Department of Public Health and Infectious Diseases, Sapienza University, 00185 Rome, Italy; 3Department of Obstetrics and Gynaecology, Fondazione Policlinico Universitario A. Gemelli IRCCS, 00168 Rome, Italy; 4Dipartimento di Scienze della Vita e Sanità Pubblica, Catholic University of the Sacred Heart, 00168 Rome, Italy

**Keywords:** COVID-19, vaccination, pregnancy, questionnaire, validation, vaccine hesitancy, communication, healthcare providers

## Abstract

Background: Fear or mistrust of the vaccine and concern for the well-being of their unborn infants are the main reasons for COVID-19 vaccine hesitancy in pregnant women. The aim of this work was to validate a questionnaire on knowledge about COVID-19 vaccination during pregnancy and to examine the sources of information in a group of new mothers, as well as their effectiveness and intelligibility. Methods: A literature review was carried out to develop a questionnaire of forty-five questions, divided into six sections, called MAMA-19. The assessment of agreement and the interrater reliability was carried out using Cronbach’s analysis and Cohen’s kappa statistic. Data obtained from the questionnaire were analysed using descriptive and univariate statistics. Results: The total alpha values in the two sections about knowledge of vaccination during pregnancy and about the effects of disease and possible post-COVID-19 consequences for the unvaccinated showed sufficient consistency, at 0.860 and 0.725, respectively. Non-vaccinated women thought that receiving the COVID-19 vaccine during pregnancy can lead to malformations in the newborn (60% vs. 40%, *p* = 0.002) and to an increased risk of foetal growth restriction (61.9% vs. 38.1%, *p* < 0.001). The percentage of vaccinated women was significantly higher than non-vaccinated when more than one professional was consulted and consistent information was received from them (74.2% vs. 25.8%, *p* = 0.008). Conclusion: The MAMA-19 questionnaire shows results in line with the literature and valid in the two main sections. It is quick to use for measuring communication effectiveness by healthcare professionals and institutions in the context of the COVID-19 vaccination campaign in the pregnant population. The results evidence that a physician’s recommendation to get vaccinated is the most important factor in maternal decision making, regardless of geographic, social or educational context.

## 1. Introduction

Pregnant women are considered a high-risk group for developing the most severe symptoms of COVID-19, due to some physiological adaptations of the immune system, cardiovascular system and respiratory system, especially if other comorbidities are present [1]. Different systematic reviews and meta-analyses report that COVID-19 vaccination in pregnant women effectively prevents the transmission of COVID-19 without significant risk of vaccine-related adverse side effects, reducing the incidence of SARS-CoV-2 infection and COVID-19-related hospitalisation among pregnant women and without any obstetrical, foetal or neonatal adverse outcomes [2,3]. The World Health Organisation (WHO), the Centres for Disease Control (CDC), the American College of Obstetricians and Gynaecologists (ACOG), the Food and Drug Administration (FDA) and multiple national immunisation advisory committees declare that pregnant women may be offered vaccination [4,5]. In Italy, the vaccination, first recommended only for pregnant women at higher risk of SARS-CoV-2 exposure or morbidity, has now been recommended by the Italian National Institute of Health (ISS) for all pregnant women, starting from the second trimester [6].

Although data suggest that vaccines are a highly effective form of protection against severe COVID-19 disease, in many countries, pregnant and postpartum women continue to receive conflicting messages, especially regarding the safety of the vaccine [7]. Even in the general population, the level of vaccine hesitancy is 10–57.8% and that of vaccine refusal is 0–24% [8]. The most common determinants affecting vaccination intention include vaccine efficacy, vaccine side effects, mistrust in healthcare, religious beliefs and trust in information sources [9,10]. Concerning the willingness of pregnant women to get the COVID-19 vaccine, several studies have reported the main reasons such as fear or mistrust of the vaccine, underestimation of the efficacy, lack of access and concern for the well-being of the unborn infant [11,12]. Lack of information about COVID-19 vaccine safety has been shown to be more common among women who are less likely to get the vaccine, demonstrating the importance of providing correct and complete data to the population. Strong recommendations by healthcare providers are one of the most important factors impacting the uptake of vaccines during pregnancy [12].

To date, several questionnaires and scales have been created to examine the impact of the COVID-19 pandemic on the pregnant population [13,14,15,16]. To the best of our knowledge, no validated questionnaire is available to measure both pregnant women’s knowledge of COVID-19 vaccination and the sources of information.

The main aims of this cross-sectional pilot study were to develop and validate a questionnaire for the evaluation of knowledge about COVID-19 infection and vaccination in a group of new mothers and to examine the sources of information used, as well as their effectiveness and intelligibility and the possible associations with vaccination status, parities, age and educational level.

## 2. Materials and Methods

The present study was performed in two main stages. The first step provided the validation of the tool, and the second step developed a cross-sectional pilot study applying the validated tool.

### 2.1. Setting

Both stages of the study enrolled women hospitalised in the postpartum unit who had just given birth. Inclusion criteria were established on the basis of the possibility to fill out a questionnaire twice and being in an not impaired emotional state. In particular, the following inclusion criteria were defined: low-risk pregnancy, non-emergency delivery, Apgar score of at least 7 at the first and fifth minutes and women over 18 years old who were able to understand Italian and without cognitive impediments. Certain factors introduce an emotional aspect in new mothers that might compromise their ability to complete the interview. Specifically, “low-risk pregnancy” is defined as a singleton, term, vertex pregnancy, and the absence of any other medical or surgical conditions. Low risk is a dynamic condition, subject to change over the course of the antepartum, intrapartum and postpartum periods. The change can be acute and unexpected [17].

High- or moderate-risk pregnant women were excluded from the pilot study because they were observed to show higher levels of anxiety than low-risk pregnancies [18]. This information was obtained from the anamnesis contained in the medical record. Even mothers with infants admitted to the neonatal pathology ward were excluded because they spent most of the day outside the obstetrics and neonatology ward and were also emotionally compromised.

### 2.2. Development and Validation of the Tool

A literature review was carried out to develop a questionnaire assessing habits and knowledge about the COVID-19 virus and vaccine. The review was carried out on December 2021 consulting PubMed and the WHO’s COVID-19 database using the following keywords: COVID-19; SARS-CoV-2; pregnancy; perinatal outcomes; vaccination; vaccination hesitancy; communication; public health. Papers were identified on perinatal outcomes caused by SARS-CoV-2 infection both in pregnant and in the overall population, and on the phenomenon of COVID-19 vaccine hesitancy among pregnant women. 

The following themes emerged: (1) pregnancy is considered an independent risk factor for adverse outcomes in case of SARS-CoV-2 infection; (2) mistrust, anxiety and scepticism of COVID-19 vaccine safety and effectiveness are the main determinants of vaccine hesitancy in pregnant women [19,20,21,22,23,24,25].

The following scales were consulted to build the questionnaire (Appendix A): Vaccination Attitudes Examination (VAX) scale [19,23], with good internal consistency, and the State Trait Anxiety Inventory (STAI) scale, an excellent scale with Cronbach’s α  =  0.93 and good test/retest reliability with an intra-class correlation coefficient of 0.80 [24,25].

The items defined for the questionnaire were discussed and formed by the expert committee composed of five professionals (three midwives, one medical doctor specialised in epidemiology and public health and one PhD in public health sciences and microbiology). The committee sorted through the questions out of the most frequently asked inquiries by the women during counselling sessions.

The developed questionnaire consists of 6 sections, with a total of 45 items.

The first section collects basic demographic and pregnancy information: age; marital status; educational qualification; nationality; profession of women and their husband/boyfriend; gravidity; parity; number of obstetrics visits during pregnancy; if the woman had been vaccinated for influenza during pregnancy; degree of apprehension about COVID-19 infection both for self and friends or relatives.

The second and third sections focus on knowledge about COVID-19 vaccination and SARS-CoV-2 infection during pregnancy, respectively.

The fourth section concerns sources of information and communication tools consulted.

The fifth section collects information about the woman’s opinion of COVID-19 vaccination.

The last section consists of five items on understanding the questionnaire and suggestions for improving it.

All the questions are closed-ended, except for the sixth section, which consists of open-ended questions. The answers to the opinion questions were given using a five-point Likert scale, and the others were dichotomised as correct or incorrect.

The first draft of the questionnaire was administered to 3 patients to report possible errors, omissions, ambiguities or misinterpretations. Minimal verbal corrections were made. 

The final version was called the MAMA-19 questionnaire (Appendix A).

It was administered twice to the women in order to examine its reliability and stability: during the delivery hospitalisation (T0) and after one day (T1).

The time window of one day (T0–T1) was chosen in consideration of the average stay of patients for delivery: 48/72 h. This time window is adequate to administer a questionnaire/interview twice in order to assess its reliability and apply an inter-rater reliability analysis (see Section 2.4).

Eligible women were informed in detail about the purpose of the study by a midwife. The women who gave their informed consent were considered enrolled in the study. To discern the T0 and T1 questionnaires, a code was given to each woman enrolled. The codes were generated and preserved by only one researcher.

### 2.3. Cross-Sectional Pilot Study

The cross-sectional study was performed according to the STrengthening the Reporting of OBservational studies in Epidemiology (STROBE) statement [26].

The study was conducted at the Teaching Hospital “Fondazione Policlinico Universitario Agostino Gemelli” in Rome, Italy. 

All women who satisfied the inclusion criteria were enrolled in consecutive order of hospitalisation on a first-come basis. Recruitable women were informed in detail about the purpose of the study by a midwife. The women who gave their informed consent were considered enrolled in this part of the study. 

The MAMA-19 questionnaire was administered to the women with an interview. Data collection was completed in February 2022. The data were entered in an Excel sheet.

### 2.4. Sample Size and Statistical Analysis

The sample size was 59 interviews for T0 and 60 for T1. The sample size was estimated considering the hypothesis that a pilot study should have a confidence level of 95% and a probability of 5% to illustrate how prevalent a problem should be in order to be important enough to want to detect it [27].

The statistical analysis was divided in two sections: one dedicated to the validation of the MAMA-19 and the second to the cross-sectional pilot study.

As regards to the validation process, demographic and other descriptive statistics of the cohort were examined with percentages and means ± standard deviation (SD); median; min; max. The assessment of agreement was carried out using Cronbach’s analysis. It was computed for each section, except for Section S1, which contains socio-demographic variables. It was computed only for T0 for the internal consistency evaluation. A Cronbach’s alpha ≥ 0.70 is generally considered an acceptable level for internal consistency [28]. Cohen’s kappa coefficient (K) is a statistic used to measure inter-rater reliability (and also intra-rater reliability) for qualitative variables. The interpretation of Cohen’s “K” was given in accordance with Landis and Koch, where a K value between 0.61 and 0.8 means substantial agreement, ≥0.8 is almost a perfect degree of agreement and =1 is perfect agreement [29].

Descriptive statistics were carried out using percentages and frequencies for qualitative variables. Mean, median, standard deviation (SD), minimum and maximum were computed for quantitative variables. 

Univariate analyses were conducted applying Chi-square and Fisher’s tests to evaluate the possible differences in the answers in the following groups: graduated and non-graduated women; women with one child vs. two or more children; aged >34 years vs. ≤34 years; COVID-19 vaccination (with at least two doses) vs. other. 

The categorical variables were dichotomised in order to apply the statistical tests in case the condition of applicability of the Chi-square tests was not met. *p* values were considered statistically significant if smaller than the significance level of α = 0.05.

The database was realised using Google Forms and exported in Excel format. All analyses were performed with SPSS (version 26.0) for Windows.

### 2.5. Ethical Issues

Prior to data collection, each participant had to read and sign an informed consent agreement, which invited them to voluntarily participate in the study in a fully confidential way, without pay, compensation or conflict of interest with the researchers. This study was carried out in accordance with the Declaration of Helsinki’s ethical standards. It is a non-pharmacological observational study and it needed no formal approval by the local ethics committee [30,31].

## 3. Results

### 3.1. Description of the Sample

Overall, 62 women were involved in the pilot study, with a total of 124 questionnaires collected at times T0 and T1. Only one patient did not fill in the T1 questionnaire because her baby was transferred to the neonatal pathology ward the day after completing the T0 questionnaire.

The 62 participants completed the MAMA-19 in 4 min, on average.

The mean age of the women enrolled was 33.37 (SD = 4.63; median = 34; min = 23; max = 46).

The socio-demographic and delivery characteristics are presented in Table 1, showing that 56.5% of the sample had a degree, while 43.5% had a lower qualification. Moreover, 39 women (62.9%) were having their first child and 23 (37.1%) had more than one child.

### 3.2. Reliability and Descriptive Analysis

Table 2 shows a summary of the Cronbach’s alpha analysis according to the topic of each questionnaire’s sections. 

Section S2 was divided for Cronbach’s alpha analysis into two groups of items: the first group refers to questions on the possible harm of the vaccine to the foetus, while the second group is on the possible benefits of the vaccine during pregnancy. The two alpha coefficients were 0.860 and 0.317, respectively.

In Section S3 on knowledge about the effects of disease and possible post-COVID-19 consequences in the unvaccinated, the total alpha value was 0.725, with sufficient internal consistency.

Section S4 reported a total alpha value = 0.228, with low consistency.

In Section S5, Cronbach’s alpha was not calculated because variables were constant.

Cohen’s kappa on agreement between T0 and T1 answers reported in all sections was substantial to perfect agreement (Table 3, Table 4, Table 5 and Table 6).

In particular, Table 3 shows the K values of Section S2. In this section, K values indicating a substantial agreement were found for questions S2.1A (K = 0.72) and 2.6 (K = 0.69); for question S2.1B, Cohen’s kappa coefficient could not be computed because at time T1, nobody chose the value 4 on the Likert scale and the symmetric two-way table was absent. For the items S2.1C, S2.2, S2.3 and S2.5, a K score ≥ 0.8 (almost perfect degree) was detected. Finally, a perfect agreement (K = 1) was found for item S2.4.

Table 4 shows the items of Section S3 on the knowledge about the effects of disease and possible post-COVID-19 consequences in the unvaccinated. The K coefficient could not be computed only for items 3.0–3.2 because the variable was constant at T0 and T1. For the item “excessive fatigue after physical exertion”, K = 0.519, so there was no agreement in answers at T0 and T1. For the items “excessive fatigue and exhaustion”, “wheezing and breathing difficulties”, “loss of sense of smell and taste”, “joint pain”, “abnormal sweating”, “chest pain” and “nausea and vomiting”, a K value between 0.61 and 0.79 was calculated. Finally, K ≥ 0.8 was calculated for items “headaches”, “attention and memory disorders”, “abnormal hair loss”, “cough” and “memory loss”.

Table 5 shows the description of Section S4 regarding the sources of information consulted on vaccination and the impact of COVID-19 disease at time T0 and T1. The kappa coefficient could not be computed only for the items “nurse”, “pharmacist”, “ASL clinic” and “vaccination centre” because the variable was constant at T0 and T1. For all remaining questions of the section, K ≥ 0.6 was found. In particular, perfect agreement (K = 1) was computed for the item “gynaecologist” of question S4.5 and for the items “midwife” and “childbirth preparation course” of question S4.6.

In Section S5, K ≥ 0.8 was found for questions S5.1 (K = 0.97), 5.2 (K = 0.957) and S5.4 (K = 0.899). Differently, for question S5.3, substantial agreement was computed (K = 0.62) (Table 6). The total number of women who declared that they did not get vaccinated for COVID-19 was *n* = 7 at T0 and *n* = 8 at T1, and those who said they were waiting to get vaccinated was *n* = 14 at T0 and *n* = 13 at T1. The total number of women who explained their reason for not getting vaccinated was *n* = 18 at T0 and *n* = 17 at T1, and those who gave their reason for waiting was *n* = 15 at T0 and *n* = 18 at T1.

### 3.3. Prevalence Results of Pilot Study

The univariate analyses regarding the associations between socio-demographic characteristics and the items on the knowledge about the vaccine effectiveness and the health effect of the virus are explained below (data not reported in tables).

There is no significant difference in knowledge about infection and the COVID-19 vaccine between graduated and non-graduated women, except for one of the symptoms, excessive fatigue after physical activity, which graduate women were more likely to identify (*p* = 0.019). Women with a university degree take more information about the COVID-19 vaccine from institutional media than other women of the group (63.8% vs. 36.2%, *p* = 0.038). Graduated women reported using the internet (*p* = 0.046) and newspapers (*p* = 0.033) more than non-graduated. Non-graduated women ask their general practitioner for information about COVID-19 more than graduated women (55.6% vs. 44.4%, *p* = 0.025), while there is no significant difference in consulting a gynaecologist, midwife and other healthcare providers or centres. No significant differences were found between women who were vaccinated and those who were not vaccinated against COVID-19 according to educational level.

As regards to women with one child and those with more than one child, the first group was more likely to have been informed about the effects of COVID-19 disease by their gynaecologist or general practitioner compared to the second group (72.5% vs. 27.5%, *p* = 0.001). They also seem to consult their gynaecologist more than other health professionals for information about the COVID-19 vaccine compared to women with more children (68.5% vs. 31.5%, *p* = 0.043). 

No significant difference in any of the items was found between women aged >34 years and ≤34 years, except for the knowledge of one of the most common COVID-19 symptoms (abnormal sweating), which is greater among younger women (87.5% vs. 12.5%, *p* = 0.024). 

Additionally, the comparisons between vaccinated and non-vaccinated women were performed. Non-vaccinated women more than vaccinated women (60% vs. 40%) thought that receiving the COVID-19 vaccine during pregnancy can lead to malformations in the newborn (*p* = 0.002) and that there may be an increased risk of foetal growth restriction (61.9% vs. 38.1%, *p* < 0.001). More vaccinated women compared to non-vaccinated women answered that the COVID-19 vaccine reduces the risk of mortality in both the general population (74.5% vs. 25.5%, *p* = 0.004) and the pregnant population (81% vs. 19%, *p* < 0.001). There is a significant difference in knowledge of post-COVID-19 symptoms such as anosmia and ageusia (76.2% vs. 23.8%, *p* = 0.026) and excessive fatigue after physical activity (75.6% vs. 24.4%, *p* = 0.045), which is higher in the vaccinated group. The non-vaccinated group also appeared less likely to ask for information about COVID-19 disease from the gynaecologist or general practitioner compared to the vaccinated group (74.5% vs. 25.5%, *p* = 0.010). Finally, when more than one professional was consulted, the consistency between the information received was significantly greater in vaccinated women than in the non-vaccinated group (74.2% vs. 25.8%, *p* = 0.008).

## 4. Discussion

The present study examines the reliability and validity of a new questionnaire to measure both the knowledge of pregnant women and the information medium they used in the context of the COVID-19 vaccination campaign. 

Statistical analysis shows a good degree of agreement among questions in every section of the MAMA-19 questionnaire: the total alpha in the two main sections “Knowledge on vaccine during pregnancy” and “Knowledge on the effect of disease and possible post COVID-19 consequences in the unvaccinated” showed sufficient consistency, with 0.860 and 0.77, respectively. One item is an exception in the first section: “Do pregnant women not vaccinated against COVID-19 have more complications, if they are positive, than vaccinated pregnant women?” (item S2.2) is least attuned to the others. Perhaps hospitalisation could introduce different knowledge of COVID-19 vaccination. In fact, the rest of the items suggest that women’s knowledge is the same at the beginning of the hospitalisation and after three days (around the end of the hospitalisation). Another aspect is that when data collection was carried out, the scientific results on serious complications in vaccinated pregnant women compared to unvaccinated pregnant women with COVID-19 infection had just been published [32], and the reports maybe were not yet well disseminated among healthcare workers and the general population. Furthermore, studies prior to the publication of the updated Italian recommendations highlighted no firm literature data related to COVID-19 vaccination during pregnancy [33,34].

Furthermore, in the section on “Knowledge on the effect of disease and possible post COVID-19 consequences in the un-vaccinated”, there was a good degree of agreement for each item, except for “excessive fatigue after physical activity”, for which a K value of less than 0.61 was calculated. Overall, these items are reliable to evaluate knowledge about SARS-CoV-2 infection.

In Section S4, Cronbach’s alpha and Cohen’s kappa analysis show that questions S4.2 (“Did you use institutional media to obtain information on COVID-19 vaccination in pregnancy?”), S4.4A (“Have you approached health professionals for information about COVID-19 vaccination in pregnancy?”) and S4.4B (“Have you approached health professionals for information about SARS-CoV-2 infection in pregnancy?”) have low internal consistency (total alpha = 0.228), but a good degree of agreement (K ≥ 0.7). It seems that these three items could not explain the sources of information consulted by women during pregnancy. Regarding questions S4.5 (“Which professionals did you turn to?”) and S4.6 (“If sufficiently satisfied with communication with professionals”), some inconsistencies were also noticed. The absolute frequency (N) of the answers for each healthcare professional/institution does not correspond between questions S4.5 and S4.6. This inconsistency is observed for all the items, so this question should be designed differently.

Question S5.1 (“Did you get vaccinated (one or more doses) for COVID-19?”) was not well understood by respondents. In fact, some of the women who said they were waiting to be vaccinated answered both question S5.3 “If no, what is the reason for not being vaccinated?” and question S5.4 “If you are waiting, what is the reason for delaying vaccination?”. On the other hand, some of the women who answered that they were not vaccinated also answered question S5.4, but they should have only answered question S5.3. Perhaps a clearer way to obtain the same information should be found.

In the second part of the study focused on the prevalence pilot study, no statistically significant differences in knowledge were found in comparison with age and educational level. In particular, the COVID-19 vaccine effect and the infectiousness of the virus, the risk of hospitalisation and mortality in the general population and in pregnant women, the possible post-COVID-19 consequences in the unvaccinated and the possible risks of the vaccine during pregnancy on the unborn child were not associated with age.

Different results were found on the basis of educational qualification in comparison with the sources of information consulted: women with a university degree seemed to gather information about the COVID-19 vaccine from different sources and instruments and consulted more institutional media than non-graduated women.

No significant differences were found between women who were vaccinated and those who were not vaccinated against COVID-19 in comparison with their education level.

The results that older age, higher education and higher income were associated with higher acceptance of COVID-19 vaccination are in agreement with several studies [35,36,37].

A systematic review published in February 2022 shows that higher-educated pregnant women were more likely to wish to have the SARS-CoV-2 vaccine. It also reports a significant level of concern against SARS-CoV-2 immunisation among the less educated. This difference might be due to the fact that more educated people have easier access to information on vaccination and are better able to understand the benefits of the vaccine [37].

The greatest concern about the COVID-19 vaccine in women who refused vaccination was a lack of data about its safety among the pregnant population; almost half of pregnant women who did not want to get the vaccine mentioned potentially harmful effects of the COVID-19 vaccine on their foetus as a reason for refusal [11]. Our analysis found that non-vaccinated women, more than vaccinated ones, thought that receiving the COVID-19 vaccine during pregnancy could lead to malformations in the newborn and to an increased risk of foetal growth restriction. These results confirm what has already been described in the literature, emphasising the importance of providing clear and easily accessible information and data so that women can make the best decision for their risk profile, guided by science rather than fear [12]. Vaccinated women, more than non-vaccinated women, answered that the COVID-19 vaccine reduces the risk of mortality in both the general population and the pregnant population; there is also a significant difference in knowledge of COVID-19 symptoms, which is higher in the vaccinated group. These findings confirm that the most important factors affecting acceptance are those related to public awareness of the risk of infection, the safety of the vaccine and the way reliable information on the need for vaccination and its safety is communicated [35].

Finally, the number of vaccinated women was significantly greater when the respondents had consulted more than one professional and there was concordance between the information received. This result suggests that receiving consistent information increases the motivation and propensity in women to get vaccinated against COVID-19, highlighting the strong influence of healthcare providers’ recommendations on the uptake of vaccines during pregnancy [12]. This also confirms that a different perception of risk and benefit exists; for this reason, it is fundamental that both healthcare providers and their patients have access to institutional, clear and up-to-date decision support material.

### Limitations

The study conducted has some limitations. On the first hand, there was a sufficient enough number of women to apply the test/retest analysis, but a selection bias could be present: the study was conducted in one region of the country, participants were recruited through convenience sampling, on a voluntary basis and with a predominantly high level of education, which may compromise external generalisability. In fact, the higher educational level (high school or bachelor’s degree) could have positively influenced the knowledge of COVID-19 vaccination during pregnancy and the use of institutional information channels and professional medical advice. The study was carried out in only one Teaching Hospital of the biggest metropolis of central Italy, and it might be interesting to propose the same questionnaire in other contexts: hospitals in the suburbs, smaller towns and different Italian regions. In these realities, the results may have been less optimistic given that two of the discriminating factors of knowledge are education level and income [36,38].

Secondly, all information is self-referenced and could introduce a measurement bias of some characteristics studied. 

Thirdly, in light of the new evidence on COVID-19 vaccination effects, the introduction of new vaccines and COVID-19’s continued viral evolution, some questions should be further modified or deleted [39]. 

Finally, the inclusion criteria, established on the basis of the ability to fill out a questionnaire twice and to exclude patients who may be emotionally compromised, are not exhaustive. This approach may have excluded a particular population of women.

## 5. Conclusions

It is pertinent to understand the impact of a vaccination campaign in general, and this also applies to COVID-19. For this reason, it is necessary to have different and valid tools to assess communication effectiveness in order to plan and monitor a vaccination campaign. It is also important to have tools that target different recipients.

With this in mind, the MAMA-19 questionnaire was designed. It gives results in line with the literature and valid in the two main sections (knowledge about vaccine during pregnancy and about the effect of disease and possible post-COVID-19 consequences in the unvaccinated). It is quick to use for measuring communication effectiveness by healthcare professionals and institutions in the context of the COVID-19 vaccination campaign in the pregnant population. In addition to monitoring the adherence and the willingness to get the COVID-19 vaccine among pregnant women, it can also be used as a basis to realise another questionnaire to assess the willingness to get another vaccine.

In relation to what has been found, the predictors of vaccine acceptance are the level of confidence in health institutions promoting vaccines and the level of awareness of COVID-19 among pregnant women, and these are largely correctable/modifiable variables [35]. There is evidence that a physician’s recommendation to get vaccinated is the most important factor in maternal decision making, regardless of geographic, social or educational context. Consistent and accurate information from healthcare providers on the current state of knowledge regarding the safety and efficacy of the vaccine as well as recommendations by scientific societies may contribute to a greater acceptance of COVID-19 vaccination among pregnant women [37]. The findings of this study, in line with the literature, suggest that health authorities should establish an immediate promotion of health programs and spread more accurate information to the population, in order to provide women and their newborns effective protection against COVID-19.

## Figures and Tables

**Table 1 ijerph-19-14886-t001:** Descriptive analysis of the sample.

Variables		*n*	%
University degree	Yes	35	56.5
	No	27	43.5
Italian nationality	Yes	57	92.0
	No	5	8.0
Employed	Yes	55	88.7
	No	7	11.3
Employed partner	Yes	61	98.4
	No	1	1.6
First child	Yes	39	62.9
	No	23	37.1
Vaginal delivery	Yes	41	66.1
	No	21	33.9
Number of obstetric visits	≤9	41	66.1
	>9	21	33.9
Gestational age (weeks)	≥37	60	96.8
	<37	2	3.2
Age group (in years)	≥34	32	51.6
	<34	30	48.4

**Table 2 ijerph-19-14886-t002:** Cronbach’s alpha analysis of the questionnaire stratified by sections.

Section	Item	Cronbach’s Alpha If Item Deleted
S2	Pregnant women who get COVID-19 vaccine risk having	
	S2.1A Neonatal malformations	0.880
	S2.1B Premature delivery	0.822
	S2.1C Intra-uterine growth restriction (IUGR)	0.883
	Total Alpha	0.860
	S2.2 Pregnant women not vaccinated against COVID-19 have the same complications, in case of a positive test result, as vaccinated pregnant women *	0.661
	S2.3 COVID-19 vaccine reduces mortality *	0.041
	S2.4 COVID-19 vaccine reduces risk of hospitalisation *	0.192
	S2.5 COVID-19 vaccine reduces short-term infectiousness *	0.190
	S2.6 COVID-19 vaccine reduces mortality during pregnancy *	0.070
	Total Alpha	0.317
S3	S3.0 Pregnant women have a higher risk of complications from COVID-19 infection than women who are not pregnant *	0.748
	S3.1 COVID-19 infection can be highly contagious *	0.728
	S3.2 Being infected with COVID-19 may require hospitalisation *	0.729
	S3.3 Possible post-COVID-19 consequences in the unvaccinated	
	Excessive tiredness and exhaustion	0.712
	Headaches	0.705
	Attention and memory disorders	0.718
	Abnormal hair loss	0.705
	Wheezing and breathing difficulties	0.721
	Cough	0.708
	Loss of sense of smell and taste	0.698
	Joint pain	0.703
	Abnormal sweating	0.703
	Chest pain	0.684
	Excessive fatigue after physical exertion	0.718
	Nausea and vomiting	0.692
	Memory loss	0.708
	Total Alpha	0.725
S4	S4.2	0.140
	S4.4A	0.230
	S4.4B	0.070
	Total Alpha	0.228

* Items dichotomised into correct and incorrect for statistical analysis.

**Table 3 ijerph-19-14886-t003:** Descriptive and reliability of “Section S2: Knowledge about COVID-19 vaccination” (*n* = 62).

Questions Section S2		T0		T1		K
		*n*	%	*n*	%	
S2.1 Pregnant women who get COVID-19 vaccine risk having					
S2.1A Neonatal malformations	Strongly disagree	41	66.1	41	66.1	**0.720**
	Disagree	10	16.1	10	16.1	
	Neutral	8	12.9	8	12.9	
	Agree	1	1.6	1	1.6	
	Strongly agree	2	3.2	2	3.2	
S2.1B Preterm birth	Strongly disagree	38	61.3	35	56.5	n.c.
	Disagree	14	22.6	17	27.4	
	Neutral	5	8.1	6	9.7	
	Agree	2	3.2	0	0.0	
	Strongly agree	3	4.8	4	6.5	
S2.1C IUGR	Strongly disagree	40	64.5	41	66.1	**0.813**
	Disagree	14	22.6	14	22.6	
	Neutral	4	6.5	3	4.8	
	Agree	1	1.6	1	1.6	
	Strongly agree	3	4.8	3	4.8	
S2.2 Pregnant women not vaccinated against COVID-19 have the same complications, in case of a positive test result, as vaccinated pregnant women ^a^	Don’t know	7	11.3	9	14.5	**0.840**
	True	20	32.3	23	37.1	
	False ^b^	35	56.5	30	48.4	
S2.3 COVID-19 vaccine reduces mortality	False	1	1.6	5	8.1	**0.814**
	True ^b^	52	83.9	52	83.9	
	Don’t know	9	14.5	5	8.1	
S2.4 COVID-19 vaccine reduces risk of hospitalisation	False	1	1.6	3	4.8	**1.000**
	True ^b^	59	95.2	58	93.5	
	Don’t know	2	3.2	1	1.6	
S2.5 COVID-19 vaccine reduces short-term infectiousness	False	21	33.9	17	27.4	**0.805**
	True ^b^	34	54.8	37	59.7	
	Don’t know	7	11.3	8	12.9	
S2.6 COVID-19 vaccine reduces mortality during pregnancy	False	3	4.8	4	6.5	**0.690**
	True ^b^	42	67.7	43	69.4	
	Don’t know	17	27.4	15	24.2	

K: agreement between T0 and T1 (0.61–0.8 substantial agreement; 0.8–0.99 almost perfect degree of agreement; 1 perfect agreement); *n* = total number of respondents; ^a^: item dichotomised into correct and incorrect; ^b^: correct answer; bold: *p* < 0.001.

**Table 4 ijerph-19-14886-t004:** Descriptive and reliability of “Section S3: Knowledge about effect of disease and possible post COVID-19 consequences in the unvaccinated” (*n* = 62).

Questions Section S3		T0		T1		K
		*n*	%	*n*	%	
S3.0 Pregnant women have a higher risk of complications from COVID-19 infection than women who are not pregnant ^a^	Don’t know	7	11.3	7	11.3	**0.805**
	Same risk ^b^	11	17.7	11	17.7	
	True	37	59.7	37	59.7	
	Partial true, only in certain cases	7	11.3	7	11.3	
S3.1 COVID-19 infection can be highly contagious	Yes	61	98.4	62	100	n.c.
	I don’t know	1	1.6	0	0	
S3.2 Being infected with COVID-19 may require hospitalisation	Yes, in certain cases ^b^	60	96.8	62	100	n.c.
	I don’t know	1	1.6	0	0	
	Yes, always	1	1.6	0	0	
S3.3 Possible post-COVID-19 consequences in the unvaccinated						
Excessive tiredness and fatigue ^c^		50	80.6	51	82.3	**0.627**
Headache ^c^		31	50.0	31	50.0	**0.807**
Attention and memory disorders ^c^		13	21.0	14	22.6	**0.858**
Abnormal hair loss ^c^		7	11.3	8	12.9	**0.924**
Shortness of breath and breathing difficulties ^c^		55	88.7	56	90.3	**0.742**
Cough ^c^		43	69.4	44	71.0	**0.884**
Anosmia and ageusia ^c^		43	69.4	43	69.4	**0.772**
Joint pain ^c^		33	53.2	36	58.1	**0.772**
Abnormal sweating ^c^		8	12.9	8	12.9	**0.713**
Chest pain ^c^		27	43.5	25	40.3	**0.735**
Excessive fatigue after physical activity ^c^		42	67.7	48	77.4	**0.519**
Nausea and vomit ^c^		13	21.0	10	16.1	**0.628**
Memory loss ^c^		7	11.3	7	11.3	**0.840**

K: agreement level between T0 and T1 (K ∈ 0.61–0.8 substantial agreement; 0.8–0.99 almost perfect degree of agreement; 1 perfect agreement); *n* = total number of responders; ^a^: item dichotomised into correct and incorrect; ^b^ correct answer; ^c^: data refers only to answer “yes”; n.c.: not computable; bold: *p* < 0.001.

**Table 5 ijerph-19-14886-t005:** Descriptive and reliability of “Section S4: Sources of information used on vaccination and impact of COVID-19 disease” (*n* = 62).

Questions Section S4		T0		T1		K
		*n*	%	*n*	%	
S4.1 I received information on the effects of COVID-19 disease during pregnancy from: ^f^						
Health personnel ^a^		51	82.3	55	88.7	**0.742 ***
No health personnel ^b^		7	11.3	7	11.3	**0.678 ***
Myself ^c^		30	48.4	27	43.5	**0708 ***
S4.2 I have used institutional media to get information about COVID-19 vaccination during pregnancy.	Yes	47	75.8	48	77.4	**0.775 ***
	No	15	24.2	14	22.6	
S4.3 Institutional media used to obtain information on COVID-19 vaccination during pregnancy: ^f^						
Internet		42	68.9	42	68.9	**0.924 ***
TV		24	39.3	27	44.3	**0.698 ***
Newspapers		10	16.4	12	19.7	**0.889 ***
Journals/periodicals		2	3.3	4	6.6	**0.651 ***
Radio		4	6.6	3	4.9	**0.849 ***
Green number		2	3.2	3	4.8	**0.792 ***
S4.4A I have approached professionals for information about COVID-19 vaccination during pregnancy	Yes	59	95.2	60	96.8	**0.792 ***
	No	3	4.8	2	3.2	
S4.4B I have approached professionals for information about COVID-19 disease during pregnancy	Yes	51	92.7	52	91.2	**0.943 ***
	No	4	7.3	5	8.8	
S4.5 Which professionals did you turn to? ^f^						
General practitioner		36	58.1	34	54.8	**0.800 ***
Gynaecologist		54	87.1	54	87.1	**0.999 ***
Midwife		9	14.5	8	12.9	**0.932 ***
Nurse		6	9.7	4	6.5	**0.783 ***
Pharmacist		3	4.8	4	6.5	**0.849 ***
ASL ^d^		3	4.8	4	6.5	**0.849 ***
Childbirth preparation course		11	17.7	9	14.5	**0.881 ***
Vaccination centre		9	14.5	9	14.5	**0.740 ***
S4.6 Sufficiently satisfied with communication with professionals: ^e,f^						
General practitioner		29	83.8	31	91.4	**0.719 ***
Gynaecologist		49	90.7	51	92.7	**0.879 ***
Midwife		9	90	9	90	**0.999**
Nurse		6	100	4	100	**n.c.**
Pharmacist		2	100	4	100	**n.c.**
ASL ^d^		4	100	4	80	**n.c.**
Childbirth preparation course		9	90	8	88.9	**0.999**
Vaccination centre		8	88.9	9	100	**n.c.**
S4.7 I have received agreed information on vaccination (if you have consulted more than one health professional)						
Agreement		32	68.1	32	69.6	**0.998 ***
Disagreement		15	31.9	14	30.4	

*n*: responders; ^a^: my gynaecologist and/or family doctor and/or general practitioner; ^b^: friends and/or relatives and/or acquaintances; ^c^: I informed myself through internet/newspaper/television; ^d^: local health unit consultatory; ^e^: data refer only to answer “satisfied”; ^f^: multiple-choice question; K: agreement level between T0 and T1 (K ∈ 0.61–0.8 substantial agreement; 0.8–0.99 almost perfect degree of agreement; 1 perfect agreement); n.c.: not computable because the variables are constants; bold: *p* < 0.05; * *p* < 0.001.

**Table 6 ijerph-19-14886-t006:** Descriptive and reliability of “Section S5: Personal position about COVID-19 vaccination” (*n* = 62).

Questions Section S5		T0		T1		K
		*n*	%	*n*	%	
S5.1 Have you been vaccinated (two or more doses) for COVID-19?	Yes	41	66.1	41	66.1	**0.970**
	I am waiting	14	22.6	13	21	
	No	7	11.3	8	12.9	
S5.2 If yes, when did you have your first dose?	Before pregnancy	8	19	8	19	**0.957**
	I trimester	8	19	9	21.4	
	II and III trimester	26	61.9	25	59.5	
S5.3 If no, what is the reason for not being vaccinated?	Poor knowledge	4	22.2	2	11.8	**0.620**
	Fear	8	44.4	9	52.9	
	Previous COVID-19	1	5.6	0	0	
	Healthcare provider does not recommend it	5	27.8	6	35.3	
S5.4 If you are waiting, what is the reason for the delay in vaccination?	After the end of breastfeeding	5	33.3	6	33.3	**0.899**
	I am not convinced about the effectiveness of the vaccine	2	13.3	3	16.7	
	I did not know about possibility to get the vaccine during the pregnancy	1	6.7	1	5.6	
	Waiting for the term of pregnancy	7	46.7	6	33.3	
	Previous COVID-19	0	0	2	11.1	

*n* = total number of respondents; K: the agreement level between T0 and T1 (K ∈ 0.61–0.8 substantial agreement; 0.8–0.99 almost perfect degree of agreement; 1 perfect agreement); bold: *p* < 0.001.

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
