# Peer review of "COVID-19 in Pregnancy: Knowledge about the Vaccine and the Effect of the Virus. Reliability and Results of the MAMA-19 Questionnaire"

_ijerph, 2022, doi:10.3390/ijerph192214886_

Round 1

Reviewer 1 Report

Dear editor,

Thank you for the kind invitation to review this manuscript.

Methodology

- Could the authors provide more explanation on what low risk pregnancy refers to?

- What happened to patients who were moderate risk?

- How does the Apgar score affect inclusion of patients and the rationale for only including patients with Apgar score of at least 7?

- Were there any reasons why patients were only recruited at time of delivery and not before delivery

-> This is odd as I noted patients who have recently delivered may also suffer from anxiety or be emotionally compromised after a long labour. In the same vein, I noted that the authors also try to exclude patients who may be emotionally compromised.

- What is the statistical power for this study?

- Could the authors provide a translated version of the questionnaire in English

-> It will help other researchers who plan to use this question

Discussion

- How do the authors think the MAMA-19 questionnaire should be used?

- Are there any outcomes that the authors have with regards to patients who scored better on this questionnaire compared to patients who scored poorly 

Minor comments

Introduction

- The authors should include a statement on broad vaccine hesitancy rates in the general population before talking about high vaccine hesitancy in pregnant women

- Please cite the following articles. 

-> https://pubmed.ncbi.nlm.nih.gov/34452026/

-> https://pubmed.ncbi.nlm.nih.gov/34835174/

Author Response

Dear Editor and Reviewers,

Thank you for giving me the opportunity to submit a revised draft of my manuscript titled MAMa-19. We appreciate the time

and effort that you have dedicated to providing your valuable feedback on our manuscript. We are grateful to the reviewers for their insightful comments on our paper. We have been able to incorporate changes to reflect most of the suggestions provided by the reviewers. We have highlighted the changes within the manuscript.

Here is a point-by-point response to the reviewers’ comments and concerns. We hope the manuscript has been improved accordingly.

Comments from Reviewer 1: 

Methodology

- Could the authors provide more explanation on what low risk pregnancy refers to?

Response: Thanks. This point is addressed in the manuscript in the following way (See introduction Paragraph):

 “Low-risk” is defined as singleton, term, vertex pregnancies, and the absence of any other medical or surgical conditions. Low risk is a dynamic condition, one subject to change over the course of the antepartum, intrapartum, and postpartum periods. The change can be acute and unexpected. In other words, “low-risk” means that there are no active complications and that there are no maternal or fetal factors that place the pregnancy at increased risk for complications.(Page 3)

The reference added is: “Board on Children, Youth, and Families; Institute of Medicine; National Research Council. An Update on Research Issues in the Assessment of Birth Settings: Workshop Summary. Washington (DC): National Academies Press (US); 2013 Sep 23. 3, Assessment of Risk in Pregnancy. Available from: https://www.ncbi.nlm.nih.gov/books/NBK201935/”

- What happened to patients who were moderate risk?

Response: Patients who were diagnosed with moderate risk during pregnancy did not receive the questionnaire to fill in. This information was obtained from the anamnesis contained in the medical record. This point is addressed in the manuscript in the following way: “High-risk or moderate risk pregnant women were excluded from the pilot study be-cause they were observed to show higher levels of anxiety than low-risk pregnancies. This information was obtained from the anamnesis contained in the medical record.” (see page 3)

- How does the Apgar score affect inclusion of patients and the rationale for only including patients with Apgar score of at least 7?

Response: We have included only women that a have the possibility to fill-in a questionnaire without anxiety and particular emotional aspects that could be to influence their attention. We think that a mother that have a baby with APGAR<7 might be in that situation. We explained that in the text.

- Were there any reasons why patients were only recruited at time of delivery and not before delivery

Response: Thank you for pointing this out. It is an opportunistic and adequate time to test-retest procedure. During the delivery hospitalization the women stay three/ four days in hospital and it is an adequate time to administer twice a questionnaire/interview in order to assess its reliability. We revised the sentence as follows (see “Development and validation of the tool” sub-paragraph): “The time window of one day (T0-T1) was chosen in consideration of the average stay of patients for delivery: 48/72 hours. This time window is adequate time to administer twice a questionnaire/interview in order to assess its reliability and apply a inter-rater reliability analysis (see “Statistical analysis” paragraph).

-> This is odd as I noted patients who have recently delivered may also suffer from anxiety or be emotionally compromised after a long labour. In the same vein, I noted that the authors also try to exclude patients who may be emotionally compromised.

Response: We agree with this comment. We have decided to include only the women that haven’t particular emotional aspects. In order to try to control this selection we have considered a systematic way to use the APGAR score or gestational age.  We have, accordingly, revised the limits of the study and emphasized this point. We revised the sentence as follows (See the sub-paragraph “Limits” in Discussion): “Finally, the inclusion criteria established on the basis of the possibility to fill-in twice a questionnaire and to excluded patients who may be emotionally compromised are not exhaustive. This approach can have excluded a particular setting of women.

- What is the statistical power for this study?

Response: We included a paragraph on the sample size that justify the power: “The sample size of 59 interviews at T0 and of 60 at T1. It will be estimate considering the followed hypotheses a pilot study with a confidence level 95% and a probability 5% that  fill in how prevalent a problem should be in order to important enough to want to detect it (Viechtbauer W, Smits L, Kotz D, Budé L, Spigt M, Serroyen J, Crutzen R.  A simple for-mula for the calculation of sample size in pilot studies. Journal of Clinical Epidemiology 2015; 68: 1375-1379. https://doi.org/10.1016/j.jclinepi.2015.04.014)”.

- Could the authors provide a translated version of the questionnaire in English

-> It will help other researchers who plan to use this question

Response: Thanks for this observation. We included English version in a supplementary file, too.

Discussion

- How do the authors think the MAMA-19 questionnaire should be used?

Response: Thanks for your question. It is can be used to monitor the adherence and the willingness to get the COVID-19 Vaccine in the women in pregnancy, but also it can be used as basis to realize another questionnaire to assess the willingness to get another Vaccine. We added this consideration in Conclusion. The sentence added is: “In addition to monitoring the adherence and the willingness to get the COVID-19 vaccine in the women in pregnancy, it can be also used as basis to realize another questionnaire to assess the willingness to get an-other vaccine.”

- Are there any outcomes that the authors have with regards to patients who scored better on this questionnaire compared to patients who scored poorly 

Response: Thank you for this suggestion. It would have been interesting to create a score. However, in the case of our study, we wanted preliminary to explore the factors related to the informative ways that could be increase the willingness to get the COID-19 vaccine. Many thanks for the observation, it is useful for the future insights.

Minor comments

Introduction

- The authors should include a statement on broad vaccine hesitancy rates in the general population before talking about high vaccine hesitancy in pregnant women

Response: We agree with the reviewer’s assessment. Accordingly, we have reported this aspect on the willingness to get vaccine in general population with the references below.

- Please cite the following articles. 

-> https://pubmed.ncbi.nlm.nih.gov/34452026/

-> https://pubmed.ncbi.nlm.nih.gov/34835174/

Response: Thanks, we included followed the references suggested in the bibliography:

Junjie Aw, Jun Jie Benjamin Seng, Sharna Si Ying Seah, Lian Leng Low. COVID-19 Vaccine Hesitancy-A Scoping Review of Literature in High-Income Countries. Vaccines (Basel). 2021 Aug; 9(8): 900.

Md. Rafiul Biswas, Mahmood Saleh Alzubaidi, Uzair Shah, Alaa A. Abd-Alrazaq, Zubair Shah. A Scoping Review to Find Out Worldwide COVID-19 Vaccine Hesitancy and Its Underlying Determinants. Vaccines (Basel). 2021 Nov; 9(11): 1243.

And also

Robinson E, Jones A, Lesser I, Daly M. International estimates of intended uptake and refusal of COVID-19 vaccines: A rapid systematic review and meta-analysis of large nationally representative samples. Vaccine 2021;39(15):2024-2034. doi: 10.1016/j.vaccine.2021.02.005.

Reviewer 2 Report

Dear Authors,

1. Introduction section, for more contributions, suggest adding related references with “mistrust, anxiety 102 and skepticism of COVID-19 vaccine safety and effectiveness.”

2. Page3, the Development, and validation of the tool, please add related references for its contribution. In addition, please expert reliability and validity.

3. Materials and methods section suggests authors explain how to collect the samples process.

4. Page 4, “The final version was called MAMA-19 questionnaire (supplementary file).” For the first time presenting MAMA-19, please present the full text. Furthermore, the supplementary file should be in English form for easy reading.

5. On Page 5, the article presented that “The socio-demographic and delivery characteristics were presented in Table 1. 56.5%” and “Table 1 shows Cronbach’s alpha analysis according to the topic of each questionnaire’s sections….” yet, I could not find the related table 1~6, all tables are missing.

Thank you.

Author Response

Dear Editor and Reviewers,
Thank you for giving me the opportunity to submit a revised draft of my manuscript titled MAMa-19. We appreciate the time
and effort that you have dedicated to providing your valuable feedback on our manuscript. We are grateful to the reviewers for their insightful comments on our paper. We have been able to incorporate changes to reflect most of the suggestions provided by the reviewers. We have highlighted the changes within the manuscript.
Here is a point-by-point response to the reviewers’ comments and concerns. We hope the manuscript has been improved accordingly.
Comments from Reviewer 2
1.    Introduction section, for more contributions, suggest adding related references with “mistrust, anxiety 102 and skepticism of COVID-19 vaccine safety and effectiveness.”
Response: Maybe the reviewer has suggested this comment in “Development, and validation of the tool”. Thanks, we added references to support our sentence:
-    Citu IM, Citu C, Gorun F, Motoc A, Gorun OM, Burlea B, Bratosin F, Tudorache E, Margan MM, Hosin S, Malita D. Determinants of COVID-19 Vaccination Hesitancy among Roma-nian Pregnant Women Vaccines (Basel). 2022; 10(2): 275. doi: 10.3390/vaccines10020275. 
-    Skjefte M, Ngirbabul M, Akeju O. et al. COVID-19 vaccine acceptance among pregnant women and mothers of young children: results of a survey in 16 countries. Eur J Epidemi-ol 2021; 36: 197–211. https://doi.org/10.1007/s10654-021-00728-6
-    Pairat, K., Phaloprakarn, C. Acceptance of COVID-19 vaccination during pregnancy among Thai pregnant women and their spouses: a prospective survey. Reprod Health 2022; 19: 74. https://doi.org/10.1186/s12978-022-01383-0
-    Murphy J, Vallières F, Bentall RP, et al. Psychological characteristics associated with COVID-19 vaccine hesitancy and resistance in Ireland and the United Kingdom. Nat Commun 12, 29 (2021). https://doi.org/10.1038/s41467-020-20226-9

2.    Page3, the Development, and validation of the tool, please add related references for its contribution. In addition, please expert reliability and validity.
Response: Thanks, we added references to support our sentence.
-    Citu IM, Citu C, Gorun F, Motoc A, Gorun OM, Burlea B, Bratosin F, Tudorache E, Margan MM, Hosin S, Malita D. Determinants of COVID-19 Vaccination Hesitancy among Roma-nian Pregnant Women Vaccines (Basel). 2022; 10(2): 275. doi: 10.3390/vaccines10020275. 
-    Skjefte M, Ngirbabul M, Akeju O. et al. COVID-19 vaccine acceptance among pregnant women and mothers of young children: results of a survey in 16 countries. Eur J Epidemi-ol 2021; 36: 197–211. https://doi.org/10.1007/s10654-021-00728-6
-    Pairat, K., Phaloprakarn, C. Acceptance of COVID-19 vaccination during pregnancy among Thai pregnant women and their spouses: a prospective survey. Reprod Health 2022; 19: 74. https://doi.org/10.1186/s12978-022-01383-0
-    Murphy J, Vallières F, Bentall RP, et al. Psychological characteristics associated with COVID-19 vaccine hesitancy and resistance in Ireland and the United Kingdom. Nat Commun 12, 29 (2021). https://doi.org/10.1038/s41467-020-20226-9

3.    Materials and methods section suggests authors explain how to collect the samples process.
Response: We have reviewed the sub-paragraph titled “Setting”. See below:
“The sample in both section of the study enrolled women hospitalized in the postpar-tum unit who have just given birth. Inclusion criteria were established on the basis of the possibility to fill-in twice a interview and to have a least possible not compromised emo-tional contest. In particular the following inclusion criteria were defined:…”
And also in the “Cross sectional pilot study”: 
“All women that satisfied the inclusion criteria were enrolled in consecutive order of hospitalization on a first come basis.”
4.    Page 4, “The final version was called MAMA-19 questionnaire (supplementary file).” For the first time presenting MAMA-19, please present the full text. Furthermore, the supplementary file should be in English form for easy reading.
Response: Thanks for this observation. We included English version as supplementary file, too.
5.    On Page 5, the article presented that “The socio-demographic and delivery characteristics were presented in Table 1. 56.5%” and “Table 1 shows Cronbach’s alpha analysis according to the topic of each questionnaire’s sections….” yet, I could not find the related table 1~6, all tables are missing.
Response: Thank you so much for catching these errors, which we have now corrected.

Round 2

Reviewer 1 Report

Nil further comments

Author Response

We have appreciated the time and effort that you have dedicated to providing your valuable feedback on our manuscript.

Thanks again.

Reviewer 2 Report

Dear Authors,

Thank you for authors revise the manuscript, but there have some issues that should explain more,

-2. Materials and methods section, The study’s tools should add related references, including reliability and validity. For more contributions, I suggest authors add each tool’s score range and a corresponding attachment to the end of the manuscript.

-table 5 was presented incorrectly.

“Table 5. Descriptive and reliability of “Sources of information used on vaccination and impact of COVID-19 diseases” section (N=62).”

For example, total samples N=62, and each subitem accounts presented over 62. Please clarify.

Page 14, lines 278-291, due to incorrect table 5; thus, the part should reword.

-Discussion section, please discussion tries to respond to studies' aims and research questions, for instance, lack of discussion via educational level.

Thank you.

Author Response

Dear Reviewer,

We appreciate the time and effort that you have dedicated to providing your valuable feedback on our manuscript. We are grateful to the reviewers for their insightful comments on our paper. We have been able to incorporate changes to reflect most of the suggestions provided by the reviewers. We have highlighted the changes within the manuscript.

Here is a point-by-point response to the reviewers’ comments and concerns. We hope the manuscript has been improved accordingly.

Thanks again.

Comments from Reviewer 2:

-2. Materials and methods section. The study’s tools should add related references, including reliability and validity. For more contributions, I suggest authors add each tool’s score range and a corresponding attachment to the end of the manuscript.

Response: Thanks. In the subparagraph “Development and validation of the tool” we have reported the construction of a new instrument. The study’s tool was developed consulting the scientific publications (see refrences numbers 19, 20, 21, 25. Accordingly, we have reported the questionnaires used from these authors that we have examined just as an inspiration for our questionnaire MAMA-19. These questionnaires are in supplemetary materials.

The followed sentence was added:

The followed questionnaires were consulted to build the questionnaire (supplementary files): Vaccination Attitudes Examination (VAX) scale [19,23] with a good internal consistency; the State Trait Anxiety Inventory (STAI) scale, it was an excellent scale with Cronbach’s α = 0.93 and a test-retest reliability good with an intra-class correlation coefficient of 0.80 [24,25].”

See page 3 lines 119-124.

-table 5 was presented incorrectly.

“Table 5. Descriptive and reliability of “Sources of information used on vaccination and impact of COVID-19 diseases” section (N=62).”

For example, total samples N=62, and each submitted accounts presented over 62. Please clarify.

Response: Thank you for pointing this out. Accordingly, in table 5 we have added foot note “f”. It explains if the item can have more than one possible answer. In this case the total may not be N=62. See pages 11-12

Page 14, lines 278-291, due to incorrect table 5; thus, the part should reword.

Response: Lines 278-291 on page 14 are concerned on table 4 “Section 3: knowledge about effect of disease and possible post COVID-19 consequences in the un-vaccinated”.

-Discussion section, please discussion tries to respond to studies' aims and research questions, for instance, lack of discussion via educational level.

Response: Thank you for pointing this out.  

We have complete the abstract according to your suggestion. Page 1

We have, accordingly, to set up the “Discussion” in order to better understand the answer to the research questions. We hope of having improved the understanding of the text.

Concerning the main aim of the study we have modified the first part to emphasize this point (lines 347-379).

Concerning the exam of the associations between the sources of information used, as well as their intelligibility with vaccination status, parities, age, educational level we have, accordingly, revised the second part of the “Discussion”, lines 392-403.

Concerning the educational level, we have reported in "Limits of the study" that our sample present an high percentage of women with high educational level, so we think that it can not be discussed adequately. (Please see lines 445-455). Additionally we haven't found  significant associations between educational level versus knowledge or the vaccinated status.